# Agent-Based Modeling of T Cell Receptor Cooperativity

**DOI:** 10.3390/ijms21186473

**Published:** 2020-09-04

**Authors:** Anastasios Siokis, Philippe A. Robert, Michael Meyer-Hermann

**Affiliations:** 1Department of Systems Immunology and Braunschweig Integrated Centre of Systems Biology, Helmholtz Centre for Infection Research, 38106 Braunschweig, Germany; anastasiossio@gmail.com (A.S.); philippe.robert@ens-lyon.org (P.A.R.); 2Institute of Biochemistry, Biotechnology and Bioinformatics, Technische Universität Braunschweig, 38106 Braunschweig, Germany

**Keywords:** TCR cooperativity, F-actin foci, immunological synapse, agent-based modeling

## Abstract

Immunological synapse (IS) formation is a key event during antigen recognition by T cells. Recent experimental evidence suggests that the affinity between T cell receptors (TCRs) and antigen is actively modulated during the early steps of TCR signaling. In this work, we used an agent-based model to study possible mechanisms for affinity modulation during IS formation. We show that, without any specific active mechanism, the observed affinity between receptors and ligands evolves over time and depends on the density of ligands of the antigen peptide presented by major histocompatibility complexes (pMHC) and TCR molecules. A comparison between the presence or absence of TCR–pMHC centrally directed flow due to F-actin coupling suggests that centripetal transport is a potential mechanism for affinity modulation. The model further suggests that the time point of affinity measurement during immune synapse formation is critical. Finally, a mathematical model of F-actin foci formation incorporated in the agent-based model shows that TCR affinity can potentially be actively modulated by positive/negative feedback of the F-actin foci on the TCR-pMHC association rate *k*_on_.

## 1. Introduction

A central process of the adaptive immune system is the recognition of foreign and unknown antigens entering the human body. T cell receptors (TCRs) on the T cell surface scan the surfaces of numerous Antigen Presenting cells (APC) in an attempt to identify processed antigen peptides presented by major histocompatibility complexes (pMHC) on the APC surface [1]. Despite the low interaction affinity between TCR and pMHC molecules [2,3,4], T cells are so sensitive that even a single pMHC can trigger cytokine production [5], and TCRs can discriminate between different pMHCs based on their association rates, *k*_on_ [6]. At the time of antigen recognition, TCRs bind to pMHC molecules to form signaling islands called TCR–pMHC microclusters. These microclusters interact with the nascent F-actin arcs in the T cell membrane and are actively transported by the centripetal flow of the arcs towards the center of the cell–cell interface [7,8,9,10]. At the later stages of antigen recognition, a characteristic pattern known as the immunological synapse (IS) emerges, where TCR–pMHC accumulate in the central region and are surrounded by adhesion complexes, such as leukocyte function-associated antigen-1 (LFA-1) bound to intercellular adhesion molecules-1 (ICAM-1) [1].

It has been observed that the avidity of microclusters is dependent on pMHC density and that, at low doses, an increase of pMHC density strengthens the avidity of the interaction. This has been interpreted as cooperativity between TCRs to boost their affinity toward pMHC during the accumulation of TCR–pMHC complexes [11]. This behavior was shown to operate on a narrow time window, where the initial TCR–pMHC ligations favor further binding events [12]. Pielak et al. [11] demonstrated that the dwell time, *τ*_off_, and consequently the dissociation rate, *k*_off_, of individual TCR-pMHC complexes remain unchanged. This led to the inference of active association rate, *k*_on_, modulation, which could be a result of a pMHC sensing multiple TCRs, since the latter are reportedly pre-organized in nanoclusters [13,14,15,16,17]. It was also shown that TCR–pMHC accumulation in the central supramolecular activation cluster (cSMAC) at the later stages of IS formation results in terminated signaling [18], suggesting that the earlier antigen recognition could result in better TCR signaling.

How is the association rate, *k*_on_, modulated though? Is there a feedback on *k*_on_? If this feedback exists, how does it originate? These important questions remain unanswered. Could it result from cortical actin cytoskeleton changes [19,20]? Is the generation and transmission of *inside-out* signals from the T cell cytoplasm towards the adhesion molecules (LFA-1) able to stabilize the T cell on the APC surface, which can consequently facilitate APC surface scanning by TCRs [21,22]? Could it be allostery that affects the way a ligand binds in the presence or absence of other binding events [23,24] or even feedback from the F-actin foci [25]?

F-actin foci are dynamically polymerized structures resulting from local nucleation of F-actin and appear at TCR–pMHC sites [25,26]. There is evidence that F-actin can bind with phospholipase C*γ*1 (PLC*γ*1), a key regulator of calcium flux downstream of TCR activation [27,28]. Interestingly, although F-actin foci depletion does not alter the IS pattern, it does manage to reduce the degree of PLC*γ*1 phosphorylation [25], suggesting that F-actin foci may locally upregulate PLC*γ*1 phosphorylation [29]. One may speculate that such a local upregulation of signaling affects the association rate of TCRs, *k*_on_, leading to the observed cooperative behavior [11].

In recent years, agent-based models are extensively used in systems biology and immunology [30,31]. These models range from cellular and molecular aggregation [32,33], receptor–ligand interactions [34,35,36], signal transduction in bacteria [37], and cellular homeostasis in diseases such as multiple sclerosis [38] to immune responses against tumors [39]. However, there are no agent-based models studying T cell signaling and consequently TCR cooperativity, to the best of our knowledge.

In order to understand whether *k*_on_ modulation is the reason behind the observed affinity modulation, we generated an agent-based model of IS formation. The model investigates the mechanisms behind molecular transport and accumulation in the different regions of the IS. Different hypotheses were tested regarding molecular transport, pMHC, as well as TCR amount titration and a possible feedback from the newly introduced F-actin foci model. The in silico experiments qualitatively reproduced the experimental results and allowed for model selection [11,12]. The model showed that pMHC or TCR titration can result in changes in the observed affinity. Furthermore, the centrally directed motion of TCR–pMHC complexes emerged as a possible mechanism, since it showed a clear modulation of the observed affinity during pMHC titration in contrast to the case of arrest of this motion. Finally, a possible positive feedback from the F-actin foci showed that a direct modulation of the TCR association rate, *k*_on_, could act as a possible mechanism.

## 2. Results

### 2.1. TCR Affinity Modulation during Antigen Recognition

Pielak et al. [11] observed that the density of pMHC molecules impacts the affinity of the interaction observed between TCR and pMHC molecules. In order to discern which kind of mechanisms could explain this property, we generated an agent-based model of the immunological synapse formation. This model takes into account the diffusion of molecules and complexes; chemical kinetics for binding and unbinding; steric exclusion of different sized complexes, called size-based segregation (SBS) modeled as a repulsive force; and finally, centrally directed flow of the complexes present in the IS, TCR–pMHC, and LFA-1–ICAM-1 due to F-actin coupling [40]. According to the experiments, the readout under investigation is the in situ dissociation rate defined as follows [11]:$${}^{cell}K_{D}=\frac{\left[TCR\mathrm{free}\right]\left[pMHC\mathrm{free}\right]}{\left[TCR-pMHC\right]}.$$

This value represents the inverse of the affinity between TCR and pMHC molecules. The dwell time between TCR–pMHC complexes has been observed to remain constant [11], suggesting that, while the dissociation rate, *k*_off_, remains invariant, the association rate, *k*_on_, of TCRs towards pMHC molecules is modulated over time.

To adhere to the experimental procedure, simulations were performed using a constant density of TCR molecules, the pMHC amount titrated between 0.07 and ≈100 molecules/µm^2^. The ^cell^
*K*_D_ observed in the simulations is shown at different pMHC densities without altering the respective association or dissociation rates, *k*_on_ and *k*_off_, of TCR–pMHC complex formation (Figure 1). Interestingly, pMHC titration resulted in a behavior qualitatively similar to the experiments, as seen in Figure 1a, while the observed IS patterns were different for the various pMHC concentrations (Figure 1b), reflecting moderate or excessive pMHC expression. ^cell^
*K*_D_ systematically declined with increasing pMHC density until a minimum value was reached, beyond which the ^cell^
*K*_D_ value increased again. The minimum value reached represents the optimum affinity between TCR and pMHC molecules.

A comparison between experimental and in silico results (Figure 1a) showed that the optimal affinity is reached for higher pMHC densities in the simulations. According to the experiments, this optimum coincides with the activation threshold for the T cell, which is the ultimate goal of IS formation. Once this goal is achieved, we could hypothesize that the behavior of TCRs might change and that the positive cooperativity can become negative. The absence of T cell activation from the model suggested that the transition from positive to negative cooperativity would happen for higher pMHCs, irrespective of the activation state of the T cell. These results further support that pMHC density plays an important role for the observed TCR–pMHC affinity.

### 2.2. TCR-pMHC Affinity Is also Affected by the TCR Density

We then inquired whether the TCR density may play a role in regulating ^cell^
*K*_D_. The model predicted that, by constricting the range of density of TCR in the synapse area from 4.5 to 55.5 molecules/µm^2^ while repeating pMHC titration (Figure 2a), ^cell^
*K*_D_ qualitatively followed the same behavior for all TCR densities checked. In all cases, ^cell^
*K*_D_ decreased with increasing pMHC density, reached a minimum value, and then increased again.

Interestingly, the minimum ^cell^
*K*_D_ shifted towards higher pMHC densities as the TCR density increased, and at the same time, the obtained minimum ^cell^
*K*_D_ values changed (Figure 2b). This suggests that the density of TCRs present in the synapse plays an important role for antigen recognition, too, and can be experimentally tested by varying the TCR amount on the T cell surface during synapse formation [41]. The results for very low pMHC concentrations, ≤1/µm^2^, showed very high ^cell^
*K*_D_ deviation from the mean, challenging their interpretation. This occurs when pMHCs become so scarce that TCRs fail to find them while searching the APC surface. Therefore, these results are not shown.

### 2.3. Centripetal Transport of TCR-pMHC as a Possible Mechanism for Affinity Modulation

The previous results were obtained without active modulation of the association, *k*_on_, or dissociation, *k*_off_, rates. All changes resulted only from the pMHC or TCR density titration, but the underlying mechanism remains to be elucidated. Does the model accommodate any unappreciated mechanism that allows for this behavior? In an attempt to shed light on why ^cell^
*K*_D_ is actively modulated during synapse formation, we decided to check the mechanisms which have a direct impact on synapse formation. One such mechanism, supported both experimentally and theoretically, is the coupling of molecules to the centripetal flow of F-actin arcs [8,9,10,40]. The strength of the model relies on its ability to perform in silico experiments with targeted inhibition of mechanisms. Therefore, the centrally directed transport of the complexes present in the IS was inhibited. The same experiments are easy to perform in the wet lab, with known F-actin inhibitors [42].

Interestingly, arrest of F-actin coupling resulted in a different qualitative behavior (Figure 3) than seen before (Figure 2). pMHC as well as TCR titration resulted in constant or, in some cases where TCR *≥* 37/µm^2^, slowly decreasing ^cell^
*K*_D_ (Appendix A). Even in the cases of decreasing ^cell^
*K*_D_, no minimum emerged. Therefore, as the model predicts, further experiments should be performed with F-actin actin arc formation inhibition [8,9,42] to investigate whether the centrally directed motion of TCR–pMHC complexes indeed supports TCR cooperativity and promotes the transition from decreasing to increasing ^cell^
*K*_D_. This behavior may be considered a transition from positive to negative feedback on TCR affinity [11]. Taken together, these results suggest that the centripetal transport of complexes due to interactions with the F-actin flow could plausibly account for TCR affinity modulation.

### 2.4. Affinity Dynamics Depend on the Time of Measurement

We then investigated the temporal evolution of affinity changes with and without interactions of complexes with F-actin (Figure 4). During the first minute of the in silico experiments, both centrally directed movement and its absence resulted in slowly decreasing ^cell^
*K*_D_ (Figure 4), in contrast to that seen in Figure 2.

However, at later time points, from 2–10 min, as the pMHC density increased, ^cell^
*K*_D_ decreased, reached a minimum value, and then increased again, recalling Figure 2. This behavior was not observed when actin coupling of molecules underwent arrest (Figure 3). Together, these results showed that the time point of affinity measurement is important, allowing for clear indication of TCR affinity modulation around 5 min after initiation of immune synapse formation. The model is in accordance with experimental findings, which suggests that peak cooperativity is achieved 4–8 min after the first observed binding event [12].

### 2.5. An F-actin Foci Model

Although the results already suggested that the centrally directed motion of TCR–pMHC complexes is a potential mechanism for TCR affinity modulation, we further investigated if active modulation of the association rate, *k*_on_, could produce the same results. According to the experimental findings, the dissociation rate, *k*_off_, remained unaffected. A possible mechanism suggested in the literature implicates the F-actin foci, which assist in TCR activation and signaling [25]. F-actin foci are dynamically polymerized structures resulting from local nucleation of F-actin and appear at TCR–pMHC sites [25,26]. We have developed an F-actin foci model (Figure 5a), where the presence of TCR–pMHC and LFA-1–CAM-1 can eventually lead to F-actin foci formation (Figure 5b; F-actin polymerization; green clusters), which in turn can affect the binding probability of TCR–pMHC complexes via a binding coefficient, *B* (Figure 5a). A detailed explanation of the model can be found in Section 4. Material and Methods.

### 2.6. Modulation of TCR-pMHC Association Rate by F-actin Foci

Now that the model incorporates the F-actin foci, we initially set out to investigate whether the observed changes in ^cell^
*K*_D_ remained as before (Figure 2 and Figure 3). Therefore, our control in silico experiment is with *B* = 1.0. In this case, the measured ^cell^
*K*_D_ values obtained in Figure 6a were the same as those shown for 18 TCR/μm^2^ in Figure 4.

We then investigated how *B* can influence this behavior. We increased *B* from *B* = 1.0 to *B* = 2.0 or 10.0. On one hand, the minimum ^cell^
*K*_D_ value reached kept on decreasing as *B* increased (Figure 6b,c and Appendix A). These results showed that the observed affinity increases in the presence of positive feedback on the TCR–pMHC association rate, *k*_on_.

On the other hand, decreasing the binding coefficient, *B* = 0.5 and 0.1 (Figure 6d,e and Appendix A), and therefore receiving a negative feedback from the F-actin foci did not alter the qualitative behavior or the minimum ^cell^
*K*_D_ value when compared to the case of *B* = 1.0 (Figure 6a and Appendix A). This happened because already formed TCR–pMHC complexes did not affect the binding probability of further TCR–pMHC complexes, resulting in a situation resembling the case of *B* = 1.0. Finally, Figure 6f (red and blue lines) shows how positive feedback affected the ^cell^
*K*_D_ value while negative feedback (Figure 6f; green and orange lines) yielded ^cell^
*K*_D_ values barely different compared to those where feedback was absent (*B* = 1.0; Figure 6f; black line).

Altogether, these results suggested that TCR–pMHC affinity can be actively modulated by positive feedback from the F-actin foci, which in turn can act as regulators of the binding probability. The feedback alone in the absence of centripetal flow of complexes failed however to reproduce the characteristic downward trend, reaching a minimum and then increasing ^cell^
*K*_D_ values dependent on pMHC density. This suggested that the centripetal transport of the complexes trumps even a strong feedback (*B* = 10.0). This analysis could be experimentally performed by inhibition of the F-actin arc and F-actin foci formation [8,9,25,42] and by comparison of the resultant ^cell^
*K*_D_ values.

## 3. Discussion

In this study, an actin nucleation model was developed, simulating the dynamics of F-actin foci formation and a possible feedback on the TCR association rate, *k*_on_. The F-actin foci model was implemented together with an agent-based model simulating the dynamics of IS formation [40]. Together, they were employed to investigate possible mechanisms resulting in T cell receptor cooperativity during antigen discrimination.

Earlier studies have shown that, during T cell activation, TCR conformational changes and an increase in avidity towards pMHC lead to improved sensing of low antigen amounts [3,43,44,45] and that even a single pMHC is able to trigger multiple TCRs and cytokine production [5]. Among possible mechanisms, TCR cross-linking can, however, be ruled out [43,46,47,48], since no physical contact between TCR–pMHC complexes was observed during the positive cooperativity phase [11]. Instead, during the formation of larger TCR–pMHC clusters, negative cooperativity was observed [11], a possible explanation being the exhaustion of these large clusters where signaling is terminated as they reach the center (cSMAC) of the IS [18].

In silico pMHC amount titration qualitatively reproduced the in vitro findings (Figure 1 and Figure 2) where the observed TCR cooperativity changed from positive to negative as the pMHC amount increased but the transition point was shifted towards higher pMHC concentrations compared to the experiments [11]. Several attempts were made in order to shift the transition point towards the experimental values. Amount titration of all the surface molecules in the IS region together with different lattice resolutions attempting to simulate closer-to-reality simulations where a single pMHC triggered a neighborhood of TCRs [5] as well as different strengths of centripetal transport failed to shift the transition point closer to [11]. Experimentally, the transition in cooperativity coincided with NFAT (nuclear factor of activated T-cells) translocation and therefore T cell activation [11]. We could hypothesize that, since the purpose of T cell activation is achieved, there is no need for more information exchange between TCR and pMHC molecules. This could lead to the anti-cooperative phase and therefore the transition point at lower pMHC concentrations than in the in silico investigation.

Pielak et al. [11] showed that a discrete level of TCR–pMHC binding events was needed in order to initiate their retrograde flow and their eventual accumulation in the cSMAC [11]. Inhibition of the retrograde flow in silico showed that the transition from positive to negative cooperativity along pMHC titration is lost (Figure 3 and Figure 4). Instead, the observed affinity remained constant or increased slightly as the pMHC amount increased. Therefore, the cooperative behavior of TCR–pMHC could result from their retrograde flow. As the initial TCR–pMHC binding events occur and start to form clusters which move towards the center of the IS, either free TCRs trapped inside these clusters can recruit more pMHC on the way or trapped free pMHC can sequentially bind to multiple TCRs, resulting in the observed increase in affinity [15,16,17]. In vitro inhibition of F-actin arc formation and therefore retrograde flow blockade [8,9] could show whether this indeed happens and the extent to which it affects the observed affinity and TCR cooperativity.

Since the cooperative behavior of TCRs was qualitatively recapitulated without active modulation of *k*_on_, we hypothesized that the missing link for shifting the transition point from positive to negative cooperativity towards the experimental pMHC amount could be the active modulation of *k*_on_, as suggested in [11]. Therefore, active modulation could be considered an allostery effect applying only to *k*_on_ [12,23], since the dissociation rate, *k*_off_, remained unchanged [11].

The F-actin foci model allowed for active modulation of *k*_on_, depending on the degree of colocalization between F-actin foci and TCRs (Figure 5). Although, we again failed to shift the transition point towards the experimental values, we did observe that a positive feedback can be responsible for increased affinity between TCR and pMHC molecules (Figure 6). Therefore, a straightforward in vitro experiment with F-actin foci inhibition [25] could shed light on the following questions: if F-actin foci indeed affected *k*_on_, to what extent would TCR affinity be modulated due to colocalization with F-actin foci and would they be responsible for positioning of the transition point at low pMHC concentrations?

It has to be noted that the model was formulated and validated based on T cell-Supported Lipid Bilayer (SLB) experimental data [11,12]. Data from 3D T cell–APC interfaces will not only allow for tuning of the model but also could help reveal the mechanisms leading to TCR cooperativity. Despite the absence of an explicit signaling model, which we are considering as a future development and is missing from the agent-based modeling field, the in silico experiments already hinted at some possible intracellular mechanisms for the observed affinity modulation during antigen recognition. Additional in silico experiments can be performed in order to understand how ^cell^
*K*_D_ and the transition point in cooperativity are affected by different TCR–pMHC association and dissociation rates. Such simulations will mimic the use of different TCR mouse model systems, such as AND and 5c.c7, and peptides, such as MCC (moth cytochrome C) and T102S, presented on MHC molecules in experimental settings [11]. The presence of costimulatory or inhibitory molecules, such as CD28 [11,29] or PD1 [49], providing positive or negative feedback on the association rate of TCR–pMHC could be introduced and tested with the model, too.

Taken together, the model showed that coupling to the F-actin retrograde flow and the consequent centripetal TCR–pMHC transport can qualitatively recapitulate the cooperative behavior of TCRs. Active modulation of the association rate, *k*_on_, by F-actin foci can increase the observed affinity but cannot be considered a leading mechanism, since the two phases of TCR cooperativity are lost by centripetal flow inhibition even in the case of strong positive feedback by the foci. The model suggested two in vitro experiments which could potentially shed light on the mechanisms behind TCR cooperativity: (1) inhibition of F-actin arc formation and consequently impaired TCR–pMHC centripetal transport [8,9] and (2) inhibition of the F-actin foci formation [25,41], alone or in combination with (1). These experiments would show whether these mechanisms can indeed affect the observed affinity of TCR molecules.

## 4. Materials and Methods

The agent-based model used in this work [40] is designed to replicate experiments performed with T cells activated on SLBs instead of APCs [1]. The model consists of two square lattices, with each node occupied by only one freely diffusing agent, namely in silico molecules. Diffusion is implemented as a random walk with a probability defined by the speed of diffusion and the simulation timestep. Binding and unbinding of molecules and/or complexes can happen only with ligands on the identical position of the two lattices, with the probabilities defined by the on and off rates, *k*_on_ and *k*_off_, of each specific molecule/complex. SBS is modeled as a repulsive force within a radius *R*_force_ between agents of different sizes, i.e., short TCR–pMHCs repel long LFA-1–ICAM-1 complexes. Finally, the centrally directed flow of complexes due to F-actin arc contraction [9] is modeled as an empirical force directed towards the center of the nascent synapse [40]. For calculation of the in situ dissociation constant, ^cell^
*K*_D_, each in silico simulation stores the density of free *TCR* and *pMHC* molecules as well as the density of *TCR–pMHC* complexes. Subsequently, ^cell^
*K*_D_ is given by calculating the fraction $\frac{\left[TCR\mathrm{free}\right]\left[pMHC\mathrm{free}\right]\;}{\left[TCR-pMHC\right]}$ [11].

For implementation of the F-actin foci model, which is derived from imaging data [25], a third square 2D lattice was created. As shown in Figure 5, TCR–pMHC complexes from the T cell lattice can create F-actin nucleation on the F-actin lattice. In this way, both lattices communicate and interact with each other. Similarly, F-actin nucleation can undergo polymerization, called F-actin nucleation polymerized, if they colocalize with LFA-1–ICAM-1 complexes. In the further presence of LFA-1–ICAM-1 in the neighborhood, defined by *R*_neighborhood_, around either F-actin nucleation or nucleation polymerized, F-actin polymerization occurs. F-actin polymerization forms small clusters representing the experimentally observed F-actin foci. Additionally, F-actin nucleation, nucleation polymerized, and polymerization inhibit themselves. The F-actin foci model should be validated by further experiments, where foci formation should be tested in the absence of ICAM-1 from the SLB.

Finally, the *binding coefficient*, *B* is the feedback from the formed F-actin foci back to the TCR–pMHC complexes. It enters the model as a coefficient multiplied with the probability of a complex to form, $p_{on}=B\frac{\tau \;}{\tau_{on}}$, where *τ* is the timestep and τ_on_ is given by $\tau_{on}=\frac{V\;*\;N_{A}}{k_{on}}$, where *V* is the volume, *N**_A_* is Avogadro’s number, and *k*_on_ the association rate of the nascent complex. *B* > 1.0 acts as positive feedback, while *B* < 1.0 acts as negative feedback on complex binding. By utilizing this feedback, the model obtained a bidirectional interaction between the complexes and the F-actin foci.

## Figures and Tables

**Figure 1 ijms-21-06473-f001:**
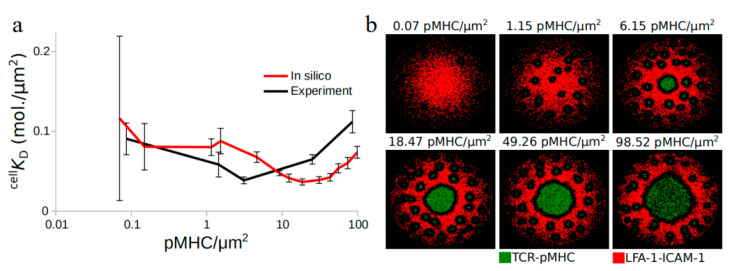
Observed T cell receptor (TCR) affinity modulation during immunological synapse (IS) formation: (**a**) in situ observed dissociation rate, ^cell^
*K*_D_, for different pMHCdensities and (**b**) immunological synapse snapshots after 10 min of contact initiation for different pMHC densities. Data were extracted from Pielak et al. [11]. TCR–pMHC: green, leukocyte function-associated antigen-1–intercellular adhesion molecules-1 (LFA-1–ICAM-1): red. Error bars represent SD of *N* = 10 simulations and SEM of at least *N* = 50 cells.

**Figure 2 ijms-21-06473-f002:**
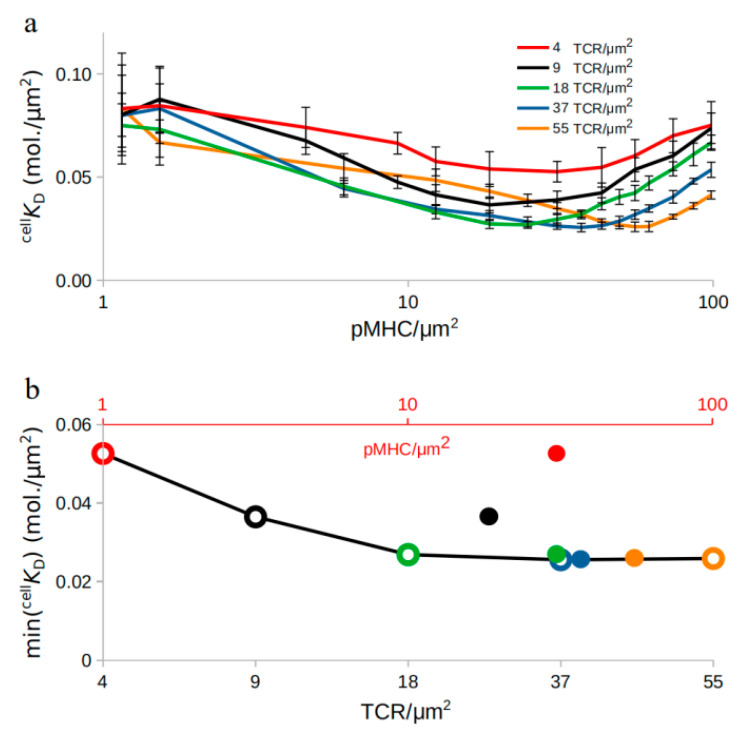
Affinity modulation during TCR amount titration: (**a**) the in situ dissociation rate, ^cell^
*K*_D_, changes for TCR amount titration. (**b**) Minimum ^cell^
*K*_D_ values for each TCR density (grey *x*-axis; circles) during TCR amount titration and pMHC density (red *x*-axis; dots) for which the minimum ^cell^
*K*_D_ values are achieved: Error bars represent SD of *n* = 10 simulations.

**Figure 3 ijms-21-06473-f003:**
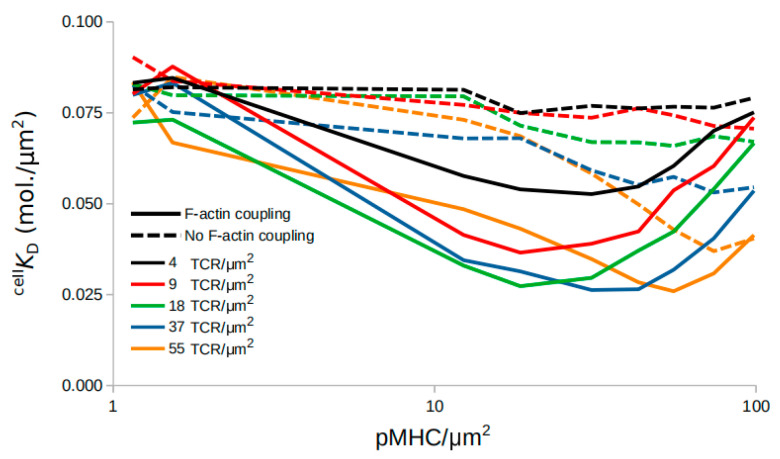
Affinity modulation during TCR and pMHC amount titration in the presence (solid lines) or absence (dashed lines) of complexes coupling to F-actin.

**Figure 4 ijms-21-06473-f004:**
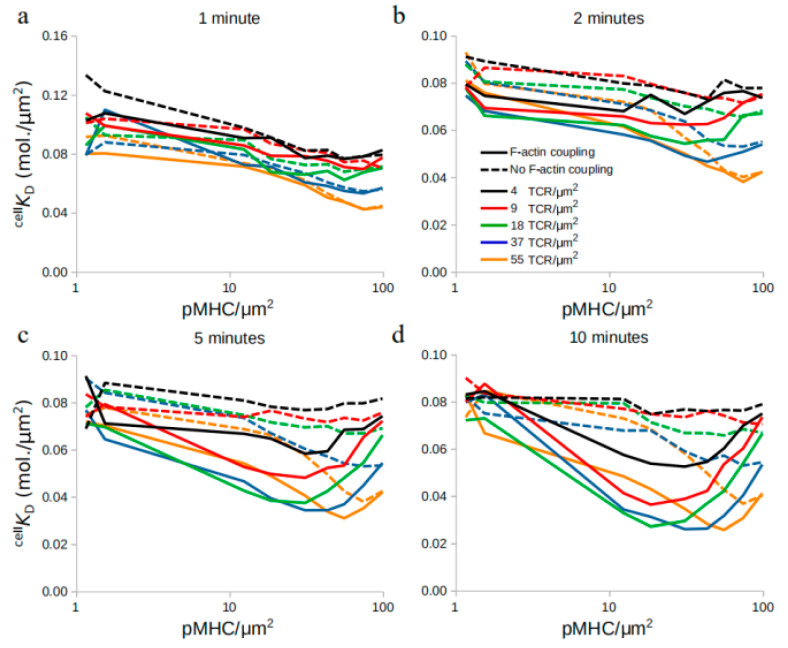
Time evolution of ^cell^
*K*_D_ during TCR and pMHC titration in the presence (solid lines) or absence (dashed lines) of F-actin centripetal transport at 1, 2, 5, and 10 min, respectively (**a**–**d**).

**Figure 5 ijms-21-06473-f005:**
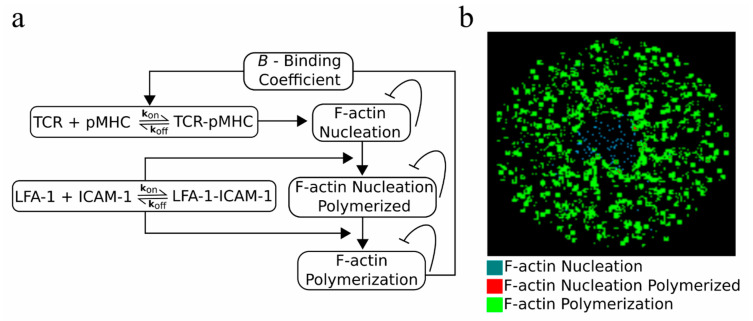
F-actin foci model: (**a**) a model of F-actin foci formation and (**b**) the formed F-actin foci during IS formation. F-actin nucleation: cyan, F-actin nucleation polymerized: red, and F-actin polymerization: green.

**Figure 6 ijms-21-06473-f006:**
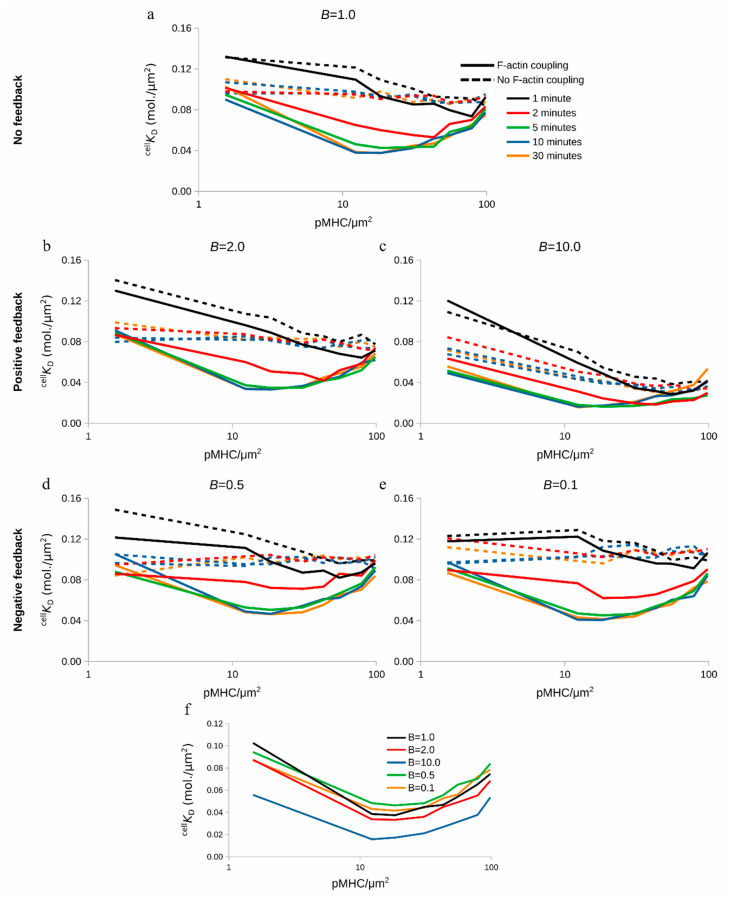
Effect of binding coefficient *B* on ^cell^
*K*_D_ in the presence (solid line) or absence (dashed line) of centripetal transport of complexes: (**a**) control case with absence of feedback, (**b**,**c**) positive feedback from the F-actin foci, (**d**,**e**) negative feedback from the F-actin foci, and (**f**) comparison of positive and negative feedback with the control case, *B* = 1.0, at 10 min of IS formation.

## Data Availability

The code was written in C++ according to all algorithmic details explained in [40] and is available on https://github.com/AnastasiosSiokis/IS_Cell/.

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
