# Peer review of "Agent-Based Modeling of T Cell Receptor Cooperativity"

_ijms, 2020, doi:10.3390/ijms21186473_

Round 1

Reviewer 1 Report

This paper presents an agent-based model for a cell to check the effect of different input parameters on the immune system. The authors have already produced a high quality journal paper on the topic, hence, high a reputation in the field.

The quality of presentation and English is very good and there is no reason the paper does not get published in this journal. 

Author Response

Dear Reviewer,

Thank you for the effort in reading our article and your comments on our previous work.

Best regards,

Anastasios Siokis

Reviewer 2 Report

The manuscript by Siokis and colleagues describes the application of an agent-based model methodology coupled with a mathematical model of F-actin foci formation for the investigation of the affinity modulation between T cell receptors (TCRs) and antigen in TCR signaling.

The article is interesting. The question is well posed and the authors show that a reasoned application of this methodology can point out new immunological insights dealing with the posed question, such as the role of the density of antigen peptide presented by major histocompatibility complexes ligands and TCR molecules. 

The abstract accurately conveys the main point and the writing is mostly acceptable.

The introduction efficiently presents the topic and the aim of the paper; however, a specific description along with the state of the art of agent-based model in immune system and in particular in T cell receptor cooperativity should be improved. It would be useful if the authors spend few words in describing these kinds of methodologies. The authors could cite some research or review articles that discuss in detail agent-based models of the immune system. There are many examples in the literature. For example: “Agent based modeling of T-reg-T-eff cross regulation in relapsing-remitting multiple sclerosis”, Pennisi, BMC Bioninformatics, 2013”.

Moreover, in the introduction section the authors mention some scientific questions that seem partially solved regarding the association rate (kon) of different pMHCs and so on. It would be valuable if authors could explain, just in few words, how they tried to address these issues through their study.

Data produced by in silico and in vitro experiments are well described and optimally produced. However, it is not clear how they intend to validate model predictions. Authors should clarify for each result obtained their validation process before to publish this work.

Results discussion is well balanced and adequately supported by the data even though the authors should more strength their results and specify any potential further investigations at the end of this section.

Furthermore, authors decided to provide their code only upon request. I strongly believe that they have to motivate and strongly justify such a choice or at least provide a temporary web graphical interface to allow reviewers and readers to launch simulations and test model applicability.

Other few small corrections are suggested below:

  1. Abstract: ligand pMHC (antigen peptide presented by major histocompatibility complexes) -> ligand of antigen peptide presented by major histocompatibility complexes (pMHC).
  2. Supplementary material: please, check and revise Figure 2 and or its caption. It seems a bit incongruent the binding coefficient definition. The authors use “BC” and “B” to indicate the same thing.
  3. Please choose only one acronym between SMAC and cSMAC.

Round 2

Reviewer 2 Report

Dear Authors,

the revisions you provided address and cover all my suggestions and objections raised. Thank you.

All the best,

Giulia Russo